# Monitoring Coastline Dynamics of Alakol Lake in Kazakhstan Using Remote Sensing Data

**Adilet Valeyev [1,2,\*], Marat Karatayev [3,4,\*], Ainagul Abitbayeva [1,2], Saule Uxukbayeva [1], Aruzhan Bektursynova [1] and Zhanerke Sharapkhanova [1]**

1    Kazakhstan Research Institute of Geography, Satbayev University, Almaty 050000, Kazakhstan; abitbayeva@satbayev.university.kz (A.A.); uxukbayeva@satbayev.university.kz (S.U.); bektursynova@satbayev.university.kz (A.B.); sharapkhanova@satbayev.university.kz (Z.S.)
2    School of Geography and Environmental Sciences, Al-Farabi Kazakh National University, Almaty 050040, Kazakhstan
3    Institute of Systems Sciences, Innovation and Sustainability Research, University of Graz, Merangasse 18-1, A-8010 Graz, Austria
4    Water-Energy-Food Nexus Partnership Program, ETH Zurich-Swiss Federal Institute of Technology, 8092 Zurich, Switzerland
\*    Correspondence: valeyev@satbayev.university.kz (A.V.); marat.Karatayev@uni-graz.at (M.K.)

**Abstract:** Alakol Lake is one of the largest hydrologically closed lake located in Balkash-Alakol River Basin in southeast Kazakhstan. Having a coastline approximately at 490 km, Alakol Lake has faced multiple threats due to both natural and anthropogenic factors as a result of tectonic movements, geology, wind-wave conditions, growing tourism activities, fishing, and transport, etc. The present study aims to investigate the historical trends in coastline changes along Alakol Lake in Kazakhstan and estimate its change rate by using remote sensing data in particular scale-space images Landsat-5 TM, 7 ETM+, 8 OLI, and Sentinel-2A. Based on Landsat and Sentinel data, the modified normalized difference water index was calculated to demonstrate the coastline changes along Alakol Lake between 1990 and 2018. Moreover, the monitoring and analysis of coastline dynamics is based on the main morphometric characteristics of Alakol Lake including water surface area, coastline length, geomorphology of the coast, etc. Our results reveal that there is a continuous coastline retreat, depending on the coast types. For example, in the case of the denudation coasts, a land inundation was from 120 to 270 m between 1990 and 2018. In the case of the accumulative coast (mainly northeast, north, and northwest coasts) a land inundation was from 200 to 900 m. A vast area of agricultural land around Alakol Lake become flooded and lost. This study demonstrates the importance of monitoring coastline dynamics because it provides essential information for understanding the coastal response to contemporary nature and anthropogenic impacts.

**Keywords:** water resources; coastline dynamics; Alakol Lake; Kazakhstan

## 1. Introduction

Most of the territory of Kazakhstan is located in a desert and semi-desert zone with a strongly continental climate [1]. The main part of the territory belongs to the closed river basins of Kazakhstan and Central Asia [2], while the northern and eastern regions of Kazakhstan belong to the basin of the Arctic Ocean [3]. The water resources in continental Kazakhstan are characterized by a high degree of inter-annual variability of surface water bodies [4]. It was shown that the spatial distribution and inter-annual variability of surface water bodies in the continental climate of Kazakhstan are affected by the impacts of climatic factors [5]. Precipitation and temperature are two dominant climatic factors that affect the spatial distribution and changes in water mass [6]. In many areas, climate change is

likely to impact on decreasing trends in water mass [7]. For example, Kazakhstan has experienced statistically increasing trends of air temperature for all seasons, while a decreasing trend of annual precipitation was observed [8]; it was predicted that the increase of average temperature and reduction in precipitation in the area around the Caspian Sea could lead to a long-term decline in the water level [9]. However, in some areas, climatic factors could lead to increasing trends of total water surface area [10].

In addition to the climatic factors, various anthropogenic activities were found related to the change of water bodies, including agricultural irrigation, dam construction, and water withdrawals for energy purposes. The expansion of irrigation over the past decades in Central Asian countries has led to a retreat of the shores and a reduction in the surface of the Aral Sea [11]. Due to the construction of thirty water reservoirs for agricultural irrigation in the basin of Ebinur Lake, the area of the lake decreased by 31.4% [12]. The excessive use of surface water for the irrigation of agricultural areas led to a reduction in the area of Lake Urmiya by approximately 88% [13]. Furthermore, future changes in the energy system of Kazakhstan are likely to lead to significant water availability in Kazakh provinces that are highly dependent on transboundary water resources from China and Uzbekistan [14]. Around 45% of available water resources originate from transboundary inflows, which are projected to decrease due to human activity by as much as 30% by 2030 [15].

Alakol Lake is one of the inland water bodies with a dynamically changing surface area and an unstable coastal zone. Alakol Lake is located at an altitude of 350 m in the southeast of the eastern province of Kazakhstan (46°04′52″ N latitude and 81°45′51″ E longitude), with an area of 3033.2 km$^2$ in the Balkhash-Alakol lowland between mountain systems of Zhetysu Alatau in the south, Tarbagatai in the north, and Barlyk in the east [16]. The main morphometric characteristics of Alakol Lake are presented in Table 1 based on the materials of Filonets and Omarov [17] and Sentinel-2B satellite images (resolution 10 m) [18] for 1973 and 2018, respectively. Several springs and streams (Urzhar, Yemel, Katynsu, Zhaman Otkel, Yrgaity, and Zhamanty), both perennial and seasonal, drain into the lake [19]. The annual mean river discharge is 2.0 km$^3$ y$^{-1}$ [19]. The climate on Alakol Lake is a continental climate, with very cold snowy days in winter but very hot and dry weather in summer. Mean annual temperature and precipitation are 6.2–7.2 °C and 152.9 mm (Tables S1 and S2—Supplementary material) [6]. The maximum and average depth of the lake is 54 and 22 m [19].

**Table 1.** Morphometric characteristics [17,18]

| Morphometric Characteristics | 1973 | 2018 |
|---|---|---|
| Lake area, km$^2$ | 2650 | 3033.2 |
| Length, km | 104 | 104 |
| Width, km | 52 | 53.5 |
| Maximum depth, m | 54.0 | - |
| Volume of water mass, million m$^3$ | 58,560 | 64,517 |
| Coastline length, km | 384.0 | 517.5 |
| Water-level (absolute m) | 347.3 | 351.1 |
| Water catchment area, km$^2$ | 47,859 | - |

The study of the level regime of Alakol Lake from 1884 to 1960 revealed the synchronous nature of fluctuations in the water level of the lake, repeating the secular and long-term trend of climate forcing [20]. Long-term fluctuations in the level of Lake Alakol can reach 5–6 m, which affect the change of the water surface area and the position of the lake coastline [19]. At the same time, it is necessary to take into account morphometric features of the water body and regional topographic profile. The lake surface area did not have significant changes over the study period, but a slight decrease in the area from 1975 to 1990 was recorded [21]. Since 1990, the trend reflects an increase in the lake area. The authors associate these processes with the consequences of global climate change in Central Asia and human activities [22].

It is necessary to study the changes in the position of the coastline in the time interval in order to understand the ongoing fluctuation processes of the Alakol Lake level. The dynamics of changes in the coastline of Alakol Lake over 28 years in four areas will be determined in the work using remote sensing data of different times. The results of the study are important for monitoring planning and developing recommendations for the organization of sustainable use of the coastal zone. The purpose of the article is to study the coastline dynamics of Alakol Lake in Kazakhstan using remote sensing data to analyze changes in the areas in the context period.

## 2. Materials and Methods

### 2.1. Satellite Images

To investigate water surface changes and to classify landscapes according to the land-forming processes, satellite images were used. Misra and Balaji [23] used remote sensing data to study the dynamics of changes in the coastline along the coastal areas of Southern Gujarat. One of the main factors for coastal erosion is the anthropogenic impacts [23]. Li et al. [24] determined the rate of change of the coasts at the center of the mouth of the river Yangtze from 1987 to 2010 using remote sensing images from different times. Qiao et al. [25] identified changes in the coastline in Shanghai Lake over the past 55 years from 1960 to 2015, using historical declassified intelligence satellite photography and Landsat time-series data at five-year intervals. Behling et al. [26] used images from a 30-year period (1984–2014) for monitoring changes in the coastline of two Namibian coastal lagoons. The method made it possible to identify the sediments and the progression of erosion on the coast in detail.

In our study, the 30 m and 10 m spatial resolution Landsat thematic mapper (TM), enhanced thematic mapper (ETM+), operational land imager and thermal infrared sensor (OLI-TIRS), sentinel 2B images (S2B) were downloaded from the United States Geological Survey for the period of 1990–2018 (Tables 2 and 3).

**Table 2.** Summary of the used datasets: Satellite images and its sensor.

| Satellite Images | Sensor |
|---|---|
| LT05_L1TP_147028_19900606_20170130_01_T1 | TM |
| LT05_L1TP_147028_19950417_20170109_01_T1 | TM |
| LE07_L1TP_147028_20000727_20170210_01_T1 | ETM |
| LE07_L1TP_147028_20050623_20170115_01_T1 | ETM |
| LE07_L1TP_147028_20100808_20161213_01_T1 | ETM |
| LC08_L1TP_147028_20150830_20170405_01_T1 | OLI TIRS |
| LC08_L1TP_147028_20180705_20180717_01_T1 | OLI TIRS |
| S2B_MSIL1C_20180729T052639_N0206_R105_T44TLN_20180729T082025 | S2B |
| S2B_MSIL1C_20180726T072159_N0206_R063_T38LNR_20180726T104944 | S2B |

**Table 3.** Satellite images with resolution, acquisition date and time.

| Satellite | Resolution | Acquisition Date | Acquisition Time |
|---|---|---|---|
| Landsat-5 | 30 m | 06/06/1990 | 04:40:26 |
| Landsat-5 | 30 m | 17/04/1995 | 04:29:06 |
| Landsat-7 | 30 m | 07/27/2000 | 05:11:26 |
| Landsat-7 | 30 m | 23/06/2005 | 05:09:40 |
| Landsat-7 | 30 m | 08/08/2010 | 05:12:27 |
| Landsat-8 | 30 m | 08/30/2015 | 05:20:03 |
| Landsat-8 | 30 m | 07/05/2018 | 05:19:20 |
| Sentinel 2B | 10 m | 29/07/2018 | 05:36:55 |
| Sentinel 2B | 10 m | 29/07/2018 | 05:36:55 |

## 2.2. Morphometric Conditions

Morphometric conditions (slope, exposure, absolute heights) of the surface part of the coastal zone were analyzed using a digital elevation model (DEM), shuttle radar topography mission (SRTM), medium-scale topographic map, and thematic maps from National Atlas of the Republic of Kazakhstan. The natural and anthropogenic conditions of the coast (tourist and recreational use, wetlands) were considered. The analysis of morphometry and natural-economic conditions made it possible to identify four key coastal sites for monitoring (Figure 1): south-western, denudation (residential territory of the Koktuma village); eastern, denudation-accumulative (recreational zone of the Kabanbai village); north-eastern, accumulative (interfluves of the Emel, Katynsu, and Urzhar rivers); northern, denudation-accumulative (to the south-west of the Kamyskala village).

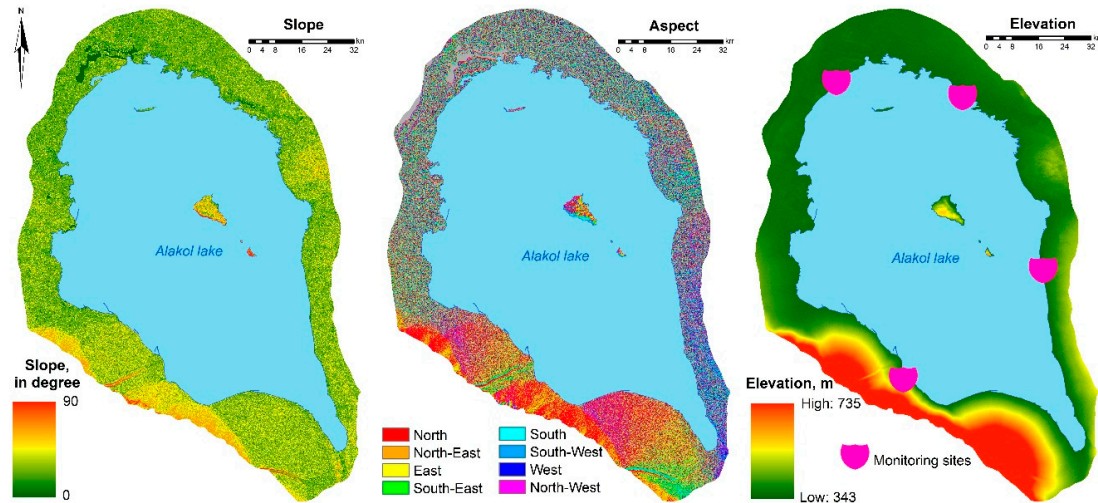

**Figure 1.** For the purpose of detailed analysis in coastline dynamics, Alakol Lake was subdivided into four monitoring sites, considering different morphometric characteristics and coast type: Monitoring site No 1 in the residential territory of Koktuma village (morphometric characteristics: south-western, denudation); monitoring site No 2 in recreational zone of the Kabanbai village (morphometric characteristics: eastern, denudation-accumulative); monitoring site No 3 in interfluves of the Emel, Katynsu and Urzhar rivers (morphometric characteristics: north-eastern, accumulative); monitoring site No 4 in Kamyskala village (morphometric characteristics: northern, denudation-accumulative. These areas were also defined based on the coast, where a higher magnitude of coastline changes were experienced in the past decades. The Alakol Lake feeding depends on rainwater, underground water, the melting of glaciers, and snow cover. The estimated living population in the area surrounded by Alakol Lake was approximately 310050 inhabitants [19]. In terms of anthropogenic activities in the Alakol Lake region, this includes settlements, farming, fishing, and tourism. The region has experienced rapid development in the last two decades, and become country's hub for summer tourism. The Alakol Lake, under natural conditions, with growing human activities, is capable of responding to climatic and non-climatic changes.

Furthermore, 20 transects from the monitoring points, located on the baseline were created on four key sites of large scale to determine the dynamics of changes in the coastline. The authors adopted the coastline for 2018 as the baseline. The positions of coastlines for 1990, 1995, 2000, 2005, 2010, 2015 and 2018 were measured from the monitoring points. Geospatial measurements and calculations of the spatial-temporal statistics were made in the ArcGIS 10.1 program.

## 2.3. Water Index

The modified index has become the commonly used water surface extraction technique from remote sensing images. Ghosh et al. [27] utilized the modified normalized difference water index

(MNDWI) algorithm for monitoring satellite images of coastline changes in Bangladesh from 1989 to 2010. Wang et al. [28] studied spatio-temporal changes in a section of Ningbo Coastline in Zhejiang Province, one of the largest port cities in China [28]. In this study, the spatio-temporal change of the Ningbo coastlines during 1976–2015 was detected and analyzed using Landsat time-series images from different sensors and the *MNDWI* were applied to discriminate surface water and land features, respectively. *MNDWI* were also used by Deus and Gloaguen [29] for the identification of water features of Lake Manyara's water surface area using a histogram segmentation technique. Feyisa et al. [30] performed water indices evaluation with Landsat 5-TM imagery of several lakes and other water bodies in different environmental conditions ranging from humid temperate through sub-tropical to tropical dry regions. In particular, the test water bodies were obtained from five different countries: Denmark, Switzerland, Ethiopia, South Africa and New Zealand. The water bodies that include small freshwater reservoirs, large lakes, harbors and the sea differ with regard to depth, turbidity, chemical composition and surface appearance. They have also demonstrated that a *MNDWI* threshold equal to 0.5 to classify open-water pixels presents an accuracy of 81%, containing inside at least 50% of detectable open water [30].

In our study, the *MNDWI* is based on the different spectral *bands*—green (*G*) and mid-infrared (*MIR*) *bands*. *Band 2* (*G*) and *band 5* (*MIR*) spectral channels are used in Landsat 5 (TM) and 7 (ETM+):

$$MDNWI = \frac{band\ 2\ (G)\ -\ band\ 5\ (MIR)}{band\ 2\ (G)\ +\ band\ 5\ (MIR)}$$

*Band 3* (*G*) and *band 6* (*MIR*) are used in Landsat 8 (OLI TIRS):

$$MDNWI = \frac{band\ 3\ (G)\ -\ band\ 6\ (MIR)}{band\ 3\ (G)\ +\ band\ 6\ (MIR)}$$

## 3. Results

Coastlines at a five-year interval for 28 years, from 1990 to 2018, were obtained using the estimated *MNDWI* index. The dynamics of the coastline of Lake Alakol was determined every 5 years since 1990, relative to the coastline of 2018. The accumulative coastal zones (mainly the northern, north-eastern, north-western parts of the coastline) undergo the greatest changes, as these areas are intensively flooded due to rising water levels in the lake.

- Monitoring site No 1 in Koktuma village: the coast has a 6-meter long coastal ledge, which is prone to intensive abrasion processes. Over a period from 1990 to 2018, the coastline has moved forward in the western direction, on average, for more than 130 m throughout the entire coast, the most drastic changes occurred during the period from 2005 to 2010 (Figure 2, Table 4).
- Monitoring site No 2 in Kabanbai village: over the 28-year period, the dynamics of the lake flooding amounted to more than 200 m (Figure 3, Table 4). The study further found that the coastal zone in Monitoring site No 1 in Koktuma village and Monitoring site No 2 in Kabanbai village is faced with myriad and growing anthropogenic pressures. These anthropogenic pressures are further fueled due to growth in human populations and density along the coast. As illustrated in Figures 2 and 3, most areas of coastal zone occupied by residential sector with recreational and agricultural activities.
- Monitoring site No 3 in the delta of Katynsu River: the coastline of the lake has moved forward towards land by approximately 1 km in some places. These are mainly low accumulative coasts, on which extensive wetlands are located (Figure 4, Table 4).
- Monitoring site No 4 in Kamyskala village: the shift of the lake's coastline is determined by an average of 120–180 m (Figure 5, Table 4).

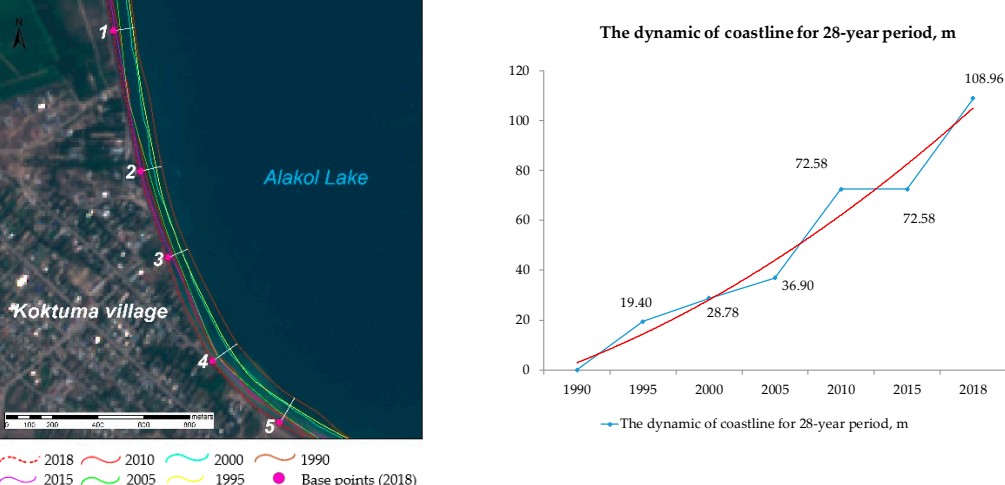

**Figure 2.** Coastline change of Alakol Lake in monitoring site No 1: south-western, denudation coast (residential area of the Koktuma village).

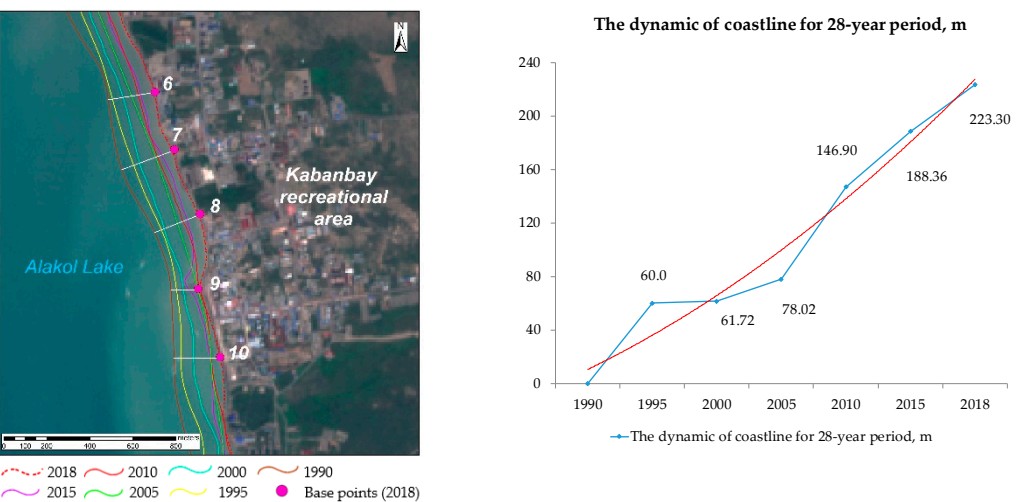

**Figure 3.** Coastline change of Alakol Lake in monitoring site No 2: eastern, denudation-accumulative coast (recreational zone of the Kabanbai village).

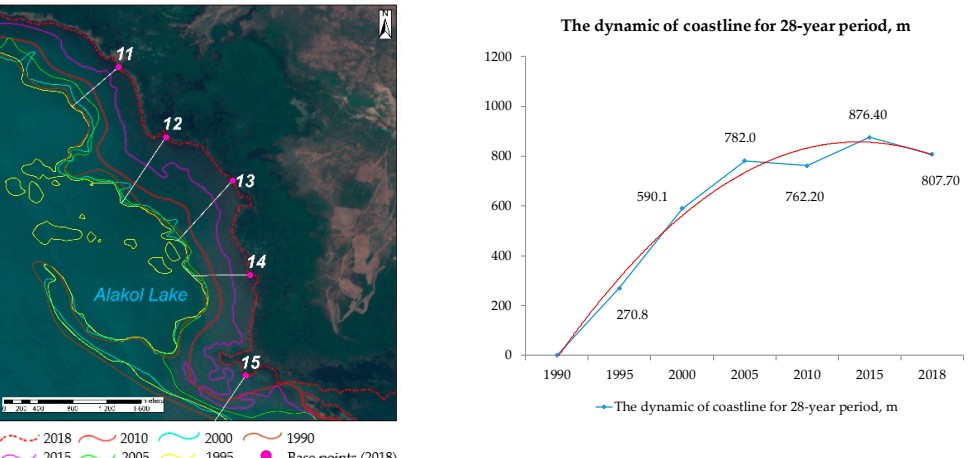

**Figure 4.** Coastline change of Alakol Lake in monitoring site No 3: north-eastern, accumulative coast (delta of the Katynsu River).

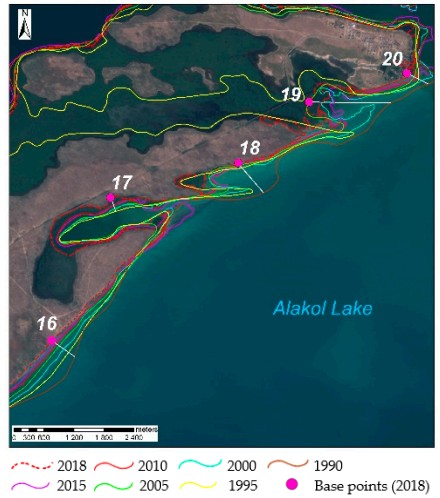
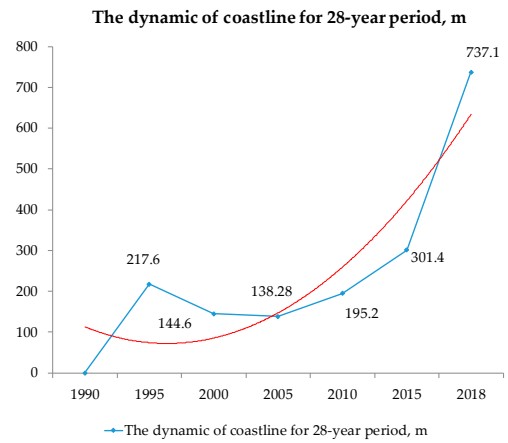

**Figure 5.** Coastline change of Alakol Lake in monitoring site № 4: northern, denudation-accumulative coast (south-west of the Kamyskala village).

**Table 4.** Dynamics of distance change from observational points to the shoreline by years, m.

| Point | 1990 | 1995 | 2000 | 2005 | 2010 | 2015 | 2018 |
|-------|------|------|------|------|------|------|------|
| **Monitoring Site No 1 in Koktuma Village** | | | | | | | |
| 1 | 95.4 | 85.7 | 74.9 | 33.4 | 24.2 | 8.3 | 0 |
| 2 | 94.6 | 72.0 | 56.4 | 36.7 | 21.0 | 4.6 | 0 |
| 3 | 97.4 | 68.9 | 67.2 | 39.2 | 28.7 | 20.1 | 0 |
| 4 | 130.6 | 90.3 | 81.2 | 33.7 | 33.1 | 33.0 | 0 |
| 5 | 126.8 | 91.5 | 83.2 | 41.5 | 36.9 | 33.7 | 0 |
| **Monitoring Site No 2 in Kabanbai Village** | | | | | | | |
| 6 | 226 | 206.8 | 159 | 114.7 | 78 | 68 | 0 |
| 7 | 272 | 217 | 177 | 89 | 86 | 64 | 0 |
| 8 | 259 | 218 | 182 | 121.7 | 88 | 63 | 0 |
| 9 | 145 | 118 | 89 | 20 | 15.6 | 67 | 0 |
| 10 | 217 | 182 | 127.5 | 44.7 | 41 | 38 | 0 |
| **Monitoring Site No 3 in Delta of Katynsu River** | | | | | | | |
| 11 | 654 | 722 | 641 | 529 | 386 | 134 | 0 |
| 12 | 823 | 1059 | 810 | 851 | 718 | 421 | 0 |
| 13 | 954 | 984 | 797 | 923 | 638.5 | 280 | 0 |
| 14 | 743.5 | 714 | 672 | 728 | 492 | 224 | 0 |
| 15 | 864 | 903 | 891 | 879 | 716 | 295 | 0 |
| **Monitoring Site No 4 in Kamyskala Village** | | | | | | | |
| 16 | 556 | 404 | 299 | 192.4 | 128 | 81 | 0 |
| 17 | 1200 | 292 | 270 | 267 | 156 | 151 | 0 |
| 18 | 760 | 120 | 106 | 64 | 57 | 57 | 0 |
| 19 | 830 | 435 | 61 | 15 | 251 | 550 | 0 |
| 20 | 339.5 | 256 | 240 | 153 | 131 | 249 | 0 |

Analysis of the results of the interpretation of the coastline of Lake Alakol according to the different-time satellite images made it possible to identify the increasing change in the water-surface area and shoreline length of the lake. The water surface areas of Alakol Lake increased from 2912.3 to 3033.2 between 1990 and 2018 (Figure 6a); the length of the coastline increased from 422.1 km in 1990 to 427.3 in 1995; from 434.2 in 2005 to 489.6 in 2010; from 482.4 in 2015 to 517.2 km in 2018 (Figure 6b).

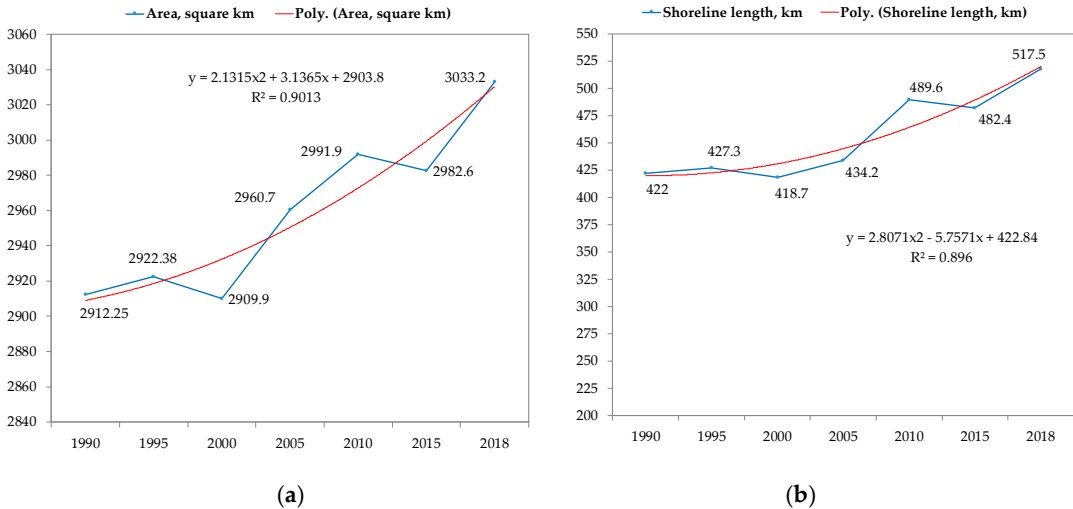

**Figure 6.** Coastline change of Alakol Lake: (**a**) Change in the surface area, km$^2$; (**b**) Change in coastline of Alakol Lake, km.

## 4. Conclusions

The use of remote sensing data allowed one to overcome the lack of ground-based information available around the Endorean water bodies of Central Asia. Moreover, the use of the DEM, Landsat, and Sentinel satellite images made it possible to conduct a detailed study of dynamics of the coasts in four key sections of the southwestern, eastern, northeastern, and northern coasts of Lake Alakol. The most critical aspect of using remote sensing data and GIS for coastal changes monitoring is the accuracy assessment. For the accuracy test, the Alakol Lake area was digitized on-screen from the Landsat images; this step is user-dependent and subjective. However, authors conducted site-visits in Alakol Lake and its monitoring points to improved knowledge on morphometric and environmental characteristics of the lake. The results showed the pattern of changes in the configuration of the coastline depending on the types of coasts. Low-lying, deltaic, northeastern, and northwestern coasts of Lake Alakol are most prone to extensive flooding. On accumulative coasts, the coastline retreat was from 204 to 925 m for a 28-year period, according to the observation points. Vast territories of agricultural land had flooded while dirt roads and engineering networks were submerged. These include the northeastern, northern, and northwestern coasts of Lake Alakol. On the denudation coasts, the dynamics of land inundation were from 66 to 218 m for the 28-year period, according to the observation points. There are several potential reasons both climatic and non-climatic for the expansion of Lake Alakol. This paper aims to show the historical trends in coastline changes and estimate the rate of change. However, it was not the aim to show mechanisms of changing lakes, factors, and linkages between them. Therefore, in further studies, we aim to focus on monitoring the hydrological, meteorological conditions, environment, and morphometry of the catchment basin.

**Supplementary Materials:** The following are available online at http://www.mdpi.com/2076-3263/9/9/404/s1, Table S1: Monthly average temperature, Table S2: Monthly, seasonal and annual rainfall in Alakol Lake.

**Author Contributions:** Conceptualization, A.V.; data curation, A.V., S.U., M.K., and A.B.; formal analysis, A.V. and S.U.; funding acquisition, A.V.; investigation, A.V., A.A., S.U., M.K., and A.B.; methodology, A.V., A.A., S.U., M.K., and A.B.; project administration, A.V.; resources, A.V. and M.K.; visualization, A.B. and Z.S.; writing – original draft, A.V., A.A., M.K., and Z.S.

**Funding:** This research was funded by Ministry of Education and Science of the Republic Kazakhstan, grant number AR05134437 "Monitoring researches on the adverse exogeodynamic processes in the coastal zone of the Alakol Lake—the territory of intensive recreational development" under the agreement No120 of March 5, 2018.

**Acknowledgments:** The authors acknowledge support from Ministry of Education and Science of the Republic Kazakhstan. Marat Karatayev would like to thank the Austrian Agency for International Cooperation in Education and Research for the award of the Ernst Mach Grant and additionally, Marat gratefully acknowledges Karl

Franzens University of Graz for providing research facilities and supporting this research. Authors would also like to thank anonymous reviewers for their helpful and constructive feedback.

**Conflicts of Interest:** The authors declare no conflict of interest.

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
