# Peer review of "Monitoring Coastline Dynamics of Alakol Lake in Kazakhstan Using Remote Sensing Data"

_geosciences, doi:10.3390/geosciences9090404_

Round 1

Reviewer 1 Report

Abstract, The term “continuous coastline shrink toward land” is inappropriate. I suggest using “coastline retreat”, "land inundation" or “lake flooding”. The sentence “As a result, vast areas of agricultural land were flooded and degraded wetlands” needs rephrasing. Why were wetlands degraded as a result of flooding? This the meaning of wetlands.   Introduction, line 37: Change “within-year” into “intra-annual”. Introduction, lines 38-40: This sentence seems fragmented. It needs rephrasing. Introduction, line 44: Change “around” into “approximately”. Introduction, line 45: “The increase in the rate of evaporation”, Please specify this increase.   Overall, paragraph 1 provides the general setting focusing on lakes’ dynamics, but mostly on lakes shrinking due to human water use and climate change. As results in Lake Alakol oppose this trend, perhaps an example of an expanding lake should be given. Line 51, Provide the lake’s altitude. Line 51, The surface area of the Lake should be corrected. More data on the rivers supplying the Lake with water must be provided (annual mean river discharge, seasonality, etc). Also, human impacts on water use and damming. Line 53. A Figure with the Lake and its hydrology (main rivers and basins) and the broader area is needed here. Table 1. Authors mention that Table 1 was based on Sentinel 2B data, but such data were not available in 1973. Therefore, only the 2018-column is based on S2B. This should be explained and corrected in text (lines 52-54) and Table. Also, why the water catchment area of the lake increased from 1973 to 2018? This might explain the expanding of its surface. Lines 76-82. These references are general works using satellite images to study coastline changes. I would suggest to omit them and provide some references on the use of MNDWI on coastline changes later in the text. Table 2. Better place the image resolution in the table. Line 100. Delete “via”. The methodology for the derivation of MNDWI is important and authors should explain it in a separate section with further analysis. Figure 1. The monitoring sites numbering should be given, to allow readers to follow the respective paragraphs. Lines 131-144. Authors should be as specific as possible explaining the processes in the study sites. Therefore, phrases as “the coastline has moved forward in the western direction” do not describe a process. This probably means inundation and flooding. The same applies to all following paragraphs. Figure 6 b. Does this graph illustrate the change of total coastline length? Line 182. Omit “are” Changes in Lake’s surface area should be related the respective changes in catchment’s area.  

Author Response

Dear Reviewer,

We are thankful for your outstanding work and for opportunity to improve this manuscript and make it stronger. We accepted all your comments and suggestions, and we acknowledged your work in our manuscript. Now we addressed your comments, please find attached file.

Authors.

Reviewer 2 Report

The results from monitoring the coastline dynamics of Alakol Lake are described in a convincing way. But three important aspects are not discussed at all in the article:

No uncertainties are given. The reader needs at least an estimate of the accuracy of the measurements in order to judge whether the surface area of Alakol Lake did really increase.  The authors monitored only four key coastal sites. They should explain what fraction they represent of the whole coastline of the lake, and what changes they expect in the part of the coastline that they did not study in detail.  The authors claim an increase of 14% for the surface area of Alakol Lake between 1973 and 2018. This is very different from the change in area of all nine lakes in central Asia considered by Bai et al. (2011). What makes Alakol Lake so special?

Here are additional suggestions for minor changes.

Lines 38 - 40 and 50 - 52: These sentence are incomplete. A verb is missing in both of them.

Line 51: The area is not 104x53.5 km= 5564 km2, but 3033.2 km2.

Line 59: 300,000 people live in the lake?

Line 63: Reference 14 did not study the lake from 1884 through 1960, but from 1975 through 2007.

Figure 1: The writing is too small.

Line 134: In Fig. 2 it does not look like the most drastic changes occurred from 2000 to 2005, but rather from 2005 to 2010.

Figures 2 - 5: The different coastlines are difficult to identify if one reads the article as a printout. The lines should be thicker.

Table 4 and line 161: Both units and uncertainties are missing with the numbers.

Line 182: I do not understand the word "semi-natural". The drainage of rivers is either caused by human activities or not ("natural").

Lines 185 - 186: Exactly what tectonic movements change the water level of the lake on the time scales of a few years? 

Author Response

Dear Reviewer,

We are thankful for your outstanding work and for opportunity to improve this manuscript and make it stronger. We accepted all your comments and suggestions, and we acknowledged your work in our manuscript. Now we addressed your comments. Please see attached file. 

Thanks,

Authors

Round 2

Reviewer 2 Report

The authors have addressed each of my comments carefully and improved the manuscript. It is now suitable for publication in Geosciences. There is only one thing left to change from my point of view: Lines 183 - 185 seem out of place. Of course the section "Results" should contain experimental results, there is no need to state this expressly.